# Adaptation and Validation of the Perception of Anomie Scale in Chilean University Students

**DOI:** 10.3390/bs14030172

**Published:** 2024-02-23

**Authors:** Fredy Cea-Leiva, Sonia Salvo-Garrido, Sergio Dominguez-Lara

**Affiliations:** 1Programa de Doctorado en Ciencias Sociales, Universidad de La Frontera, Temuco 4780000, Chile; fcea@uct.cl; 2Escuela de Posgrado, Universidad Católica de Temuco, Temuco 4780000, Chile; 3Departamento de Matemática y Estadística, Universidad de La Frontera, Temuco 4780000, Chile; 4South American Center for Education and Research in Public Health, Universidad Privada Norbert Wiener, Lima 15108, Peru; sergio.dominguez@uwiener.edu.pe

**Keywords:** anomie, disintegration, deregulation, emerging adulthood, measurement

## Abstract

The Perception of Anomie Scale (PAS) is a measure used to evaluate the state of society and whether it is disintegrated and deregulated. Although widely used, the psychometric properties of reliability, validity, and measurement invariance according to sex have not been studied in the Chilean university population. To explore these properties, a cross-sectional study was carried out with 383 students from public (45.7%) and private (54.3%) universities, with a mean age of 21.5 years (SD = 3.3). A CFA and ESEM were performed, which confirmed the two-factor correlated structure, achieving better goodness-of-fit indices by eliminating five items (RMSEA = 0.09; CFI = 0.98; TLI = 0.95). This also provided evidence of reliability and measurement invariance according to sex. This study provides evidence of the psychometric quality of the PAS scale, thus allowing its use in the Chilean university context.

## 1. Introduction

Anomie is a complex concept [1] and historically presents a variety of definitions [2]. Durkheim [3] developed his theory of social change, which centered on a socially constructed morality [4,5] that regulates an individual’s behavior. The theory highlights how the division of labor disconnects the subject from tradition and their ancestors, separating individual judgment from the collective. This greater complexity or overspecialization [6] causes disorientation in social relationships and stresses morality (doctrines, customs, norms), which for this author is the basis of social integration, and anomie arises sui generis, which would disappear [5] while the discipline and morality of the new order reappear. Thus understood, anomie can be inserted into the debate on the theory of social regulation [4], where the control of individual passions occurs through the internalization of a socially constructed morality; however, when these mechanisms fail, a state of “immorality” is produced [5], the central characteristic of which is that the social regulation apparatus disappears, leaving the subject with no guidance for their behavior. As the concept is often understood, this lack of rules would occur in the transition between an old order and a new one [7,8].

This approach opens up various perspectives, highlighting the consequences of the lack of normativity or its ambiguity [9,10] at the social and personal levels. For the individual, Freud [11] mentions a tension resulting from the repression of biological urges due to social coercion, producing a malaise inherent to living in society. Thus, abnormal behaviors could be explained by the triumph of instinct over social repression. Merton [12] seeks an explanation for abnormal behavior in the social structure and culture rather, than developing a pathological individual version [5]; hence, he posits that in society, there is a breakdown between social goals, a limited availability of opportunity within a class structure, and an erosion of the admissible means to achieve those goals [12]. Cultural aspirations, especially in American society, have installed wealth and economic success as the great objectives to be achieved. Exaltation in this goal has caused demoralization and distancing from the means, especially considering that in the social structure, the distribution of opportunities is heterogeneous and benefits the upper socioeconomic strata to a greater extent [5]. Thus, anomie would occur when the end takes precedence over the legitimate means to achieve it. With Merton [12], the concept of anomie transitions to a sociology of deviant behavior and crime [2], an analysis that deepens with the developments of Messner and Rosenfeld [13], with the theory of institutional anomie [14].

In line with Durkheim’s initial contributions, the present study understands anomie as the perception that a given society has become disintegrated and deregulated. Disintegration implies a perceived breakdown in the social fabric, including a lack of trust and moral standards. Deregulation involves the perception that a society’s leadership is collapsing, that it is illegitimate and ineffective, and that leaders do not represent them and do not follow fair decision-making processes [15]. Thus, anomie is an evaluation of the state of society made by a subject (intersubjective construction) through processes of interpretation and social communication that may or may not be influenced by objective processes such as economic crises, wars, or social outbursts, and which may or may not be affected by objective processes such as economic crises, wars, social upheavals, etc. [15]. This current is separated from studies focused on anomie as a mental state of individuals in an anomic society, the axes of which are deviant behaviors, crime, as well as despair, personal confusion, frustration, meaninglessness, powerlessness, and loneliness [16,17,18,19,20].

Current evidence indicates that anomie is directly related to social unrest or social discontent [15,21,22], violent extremism [23], inequality [15,24,25,26], moral polarization [27], social control [28] and control in the context of the COVID-19 pandemic [29], greed [30], as well as higher divorce rates [31]. At the political level, it was found that the greater the deregulation (a dimension of anomie), the greater the support for proposals pertaining to protectionism and economic redistribution [32] as well as to prosocial behaviors [29], and greater support for extreme positions on the right and left [32]. Additionally, the greater the disintegration (another dimension of anomie), the greater the support for strong conservative/authoritarian style leaders [24,25,27]. There is also evidence that perceived anomie is associated with lower interpersonal trust, which would also be associated with lower political trust [33]. In turn, there is evidence that perceived anomie affects people’s representations of past events or political figures [34], and would also influence a negative bias of some decline into the future [21,22,35].

Just as there are many definitions of anomie, there are also multiple instruments for its empirical measurement. Probably the most widely used measure is Srole’s scale [16], which is an approach that addresses the subjective experiences of people exposed to the social condition of anomie; therefore, the author here calls it anomia to differentiate it from the social [20]. The original scale has five items for five different dimensions, which present the difficulty of measuring complex theoretical constructs with a single item. Several instruments are based on this measure, such as those of Fischer [36] and Teevan [37]. The latter has three dimensions: powerlessness, distrust, and social isolation.

Other scales that assess the construct are Dean’s [38] normlessness scale, which seeks to measure existential angst and religious relativism [20], and McClosky and Schaar’s [18] anomie scale based on nine dichotomous items (agree/disagree) that measure an individual’s state of mind. Elmore’s [39] anomie scale is also based on Srole’s, but incorporates more items per dimension.

More recently, the Ådnanes scale [40] has postulated three dimensions: one associated with the psychological and two of a more social nature, one based on normlessness and the other on perceptions of social change. Bjarnason [20] hypothesized an anomie scale with two dimensions: exteriority and constraint. The former points to experiencing the social world as an objective and predictable reality, while constraint refers to the tension between personal engagement with societal expectations. The scale by Heydari et al. [41] also has three dimensions: powerlessness, meaninglessness and distrust, and the fetishism of money. Bashir and Bala [42] developed a 21-item scale grouped into three dimensions: meaninglessness, distrust, and moral decline. In 2021, a scale was developed to measure organizational anomie, consisting of a single dimension and eight items and focusing on employees’ psychosociological perceptions [43]. Vilca et al. [44] developed an anomie scale for the pandemic context, defining a behavioral factor that refers to non-compliance with norms and an affective factor that refers to the degree of dissatisfaction generated by the new norms.

Without being an exhaustive review of the large number of instruments available for measuring anomie, it is important to note that most belong to the psychological current related to despair, personal confusion, frustration, meaninglessness, powerlessness, and loneliness [16,17,18,19,20], which departs from the initial meaning of the Durkheimian concept, associated with a central discussion on social integration and regulation.

This research uses the Perception of Anomie Scale (PAS) by Teymoori et al. [15], which comes from social psychology and takes up Durkheim’s [3] central concepts, the purpose of which is to assess society in relation to intersubjectivity. They build and validate a two-dimensional instrument, concerning breakdown in social fabric and breakdown of leadership. This instrument was applied in 30 countries to obtain convergent, discriminant, and predictive validity through a stepwise study [15]. Compared with other countries, Chile had an average score of 4.53 on the scale, ranking 13th among the nations with the greatest perception of anomie.

While the fit indicators measured in the original study [15] present evidence of validity supporting the two-factor correlated structure (RMSEA = 0.06; CFI = 0.96; TLI = 0.95), in addition to satisfactory internal consistency, with Cronbach’s alphas of 0.88 for the entire scale, 0.81 for perceived breakdown in the social fabric, and 0.87 for perceived breakdown of leadership, there is no version adapted to Chile.

Concerning previous instrumental studies, only one study was found that analyzes the psychometric properties of the PAS in Colombian students [45], although there are some important limitations. First, the reliability analysis was performed before the structural analysis, which indicates that the internal structure was assumed before it was evaluated. Second, the α coefficient was calculated for the complete scale, when its calculation should be done by dimensions, although at this level, breakdown of leadership had more support than breakdown in social fabric. Third, by not establishing a minimum criterion for factor loadings, there are some items whose value is too low to be considered relevant, such as items 5 (λ = 0.44) and 6 (λ = 0.30).

Chile presented high anomie values in 2016; however, in October 2019, a social movement of significant proportions was unleashed, which put Chilean institutionality at risk and involved, among other consequences, the drafting of a new constitution by a majority of left-wing constituents and the election of Gabriel Boric as president.

The pandemic and the process of drafting the new constitution moderated these pressures for change, resulting in the proposal being rejected by a large majority, giving new strength to the extreme right in particular. These pendular movements may be based on the perception of an anomic Chilean society, and therefore, it is necessary to have valid and reliable instruments to measure anomie, which is the purpose of this study.

According to the above, the general objective is to analyze the psychometric properties of the PAS, which is divided into three specific objectives: (i) to evaluate the factor structure of the scale; (ii) to evaluate its reliability; (iii) to evaluate the measurement’s invariance by sex. Three hypotheses emerge from these objectives: (H1) the PAS has a two-factor correlated structure; (H2) the reliability results of the PAS in Chilean university students are adequate; and (H3) the results of the models for men and women fit acceptably, demonstrating measurement invariance by sex.

## 2. Materials and Methods

### 2.1. Participants

The population defined for this study was university students from Temuco (N = 53,346), who contributed to a non-probability convenience sample of 383 students who freely agreed to participate in the study. The minimum sample size required (n = 200) was defined based on specific recommendations by considering effect size (0.50 as the minimum value of the factor loadings), number of factors (2) and items (12) [46]. The sample were all aged between 18 and 50 (M = 21.5 years; SD = 3.3). In total, 244 (63.7%) reported female gender, 132 (34.5%) reported male gender, and 7 (1.8%) reported another gender; 175 (45.7%) students from public universities and 208 (54.3%) students from private universities participated, belonging to programs in social sciences and humanities (16.7%), engineering (14.1%), health (31.3%), and education (37.9%).

### 2.2. Instruments

The Perception of Anomie Scale (PAS), constructed by Teymoori et al. [15], was used. The PAS measures the perception of the state of society through 12 seven-point ordinal response items ranging from strongly disagree (1) to strongly agree (7). It presents a structure of two correlated dimensions called “Breakdown in social fabric”, composed of items 1 through 6 (e.g., “People think that there are no clear moral standards to follow”) and “Breakdown of leadership”, items 7 to 12 (e.g., “The government works towards the welfare of people”). It was validated in Australian and American populations, showing good psychometric properties. The PAS has undergone no validity study in Chile. The content of the items is presented in Table 1. In addition, a sociodemographic characterization questionnaire was used, which included questions such as age, gender, and others.

### 2.3. Procedure

The guidelines of the International Test Commission [47] were followed for the adaptation and validation of the scale. After identifying the scale, contact was made with the author, who authorized its use and provided the scale in Spanish. This was used in Chile and 28 other countries in the original study, where back-translation processes were carried out to adapt it to the languages of each participating country. On that occasion, the scale was used to relate anomie to indicators of social stability to compare countries and conduct predictive validity studies.

The scale was sent to a group of Chilean experts to evaluate possible linguistic or cultural differences and/or comprehension of the items, concluding that they were sufficiently clear for use with Chilean university students. A group of students was also involved in evaluating the understanding of both the instructions and the items, and this process ended without changes in the Spanish version sent by the authors of the original scale.

For the application, students were contacted via e-mail and in person through a QR code linked to a computerized platform Question Pro (advanced version), where the instruments were placed. Informed consent forms were given to all participants to safeguard the ethical principles of the project. This study was approved by the Science Ethics Committee of the Universidad de La Frontera (File No.26_22; Research Protocol Page No.094/22).

### 2.4. Data Analysis

The analysis process was developed in stages following the proposed specific objectives (SO). To evaluate SO1, the most appropriate factor structure of the PAS was determined by assessing models with theoretical support using a confirmatory factor analysis (CFA) and exploratory structural equation modeling (ESEM). The latter option was chosen because ESEM is primarily a confirmatory technique [48,49,50]. However, it is more flexible and has fewer identification and specification errors than the CFA [48,51]. In addition, ESEM provides a more adequate representation of the data in terms of fit, particularly for confirmatory purposes [48,51]. It also accurately estimates the relationships among latent factors [52,53]. Another key aspect is that ESEM models tend to be more closely aligned with the theoretical representation of the construct that the instrument is intended to measure [54], since, in certain cases, a CFA model is also nested within an ESEM model, so the two can be compared [48]. If it fits the data better than the CFA, the estimate of the factor correlation will likely be substantially less biased and of smaller magnitude in the ESEM model than in the CFA model [48].

In that sense, implementing ESEM on theoretically complex constructs could provide a more realistic point of view, and thus, an understanding of the dimensional nature of the construct [55]. Figure 1 shows the fitted models. The one-factor model and 2 two-factor models were evaluated with a CFA. Models 3 and 4 are representations of ESEM models.

The Mplus 7.11 program [56] was used. The evaluation was performed with the weighted least square mean and variance-adjusted (WLSMV) estimation method [57], recommended for analyzing ordinal variables [58] over a wide range of sample sizes [59]. Furthermore, WLSMV makes no distributional assumptions on the observed variables [60]. Consequently, the robust standard errors of the structural coefficients are more accurate than those obtained with MLR and ULSMV in all situations of skewed data [61]. In the case of the ESEM model’s estimations, target rotation makes it possible to use this technique in a confirmatory mode, given that it produces the closest rotated solution to a pre-specified configuration of loadings [62]. In the present study, the main factor loadings were freely specified, while the secondary factor loadings were specified as close to zero (~0). This provides a more robust a priori model and facilitates the interpretation of the results [48]. Goodness-of-fit was assessed through the following indicators: comparative fit index (CFI), Tucker–Lewis index (TLI), root mean squared error of approximation (RMSEA), and standardized root mean square residual (SRMR). This last indicator was used to complement the RMSEA, as new research has shown that SRMR outperforms RMSEA when the data evaluated are categorical [63]. To assess the models, an adequate fit is assumed when the CFI and TLI are over 0.90 [64]. The RMSEA values below 0.08 are adequate [65]. With respect to the SRMR, a value below 0.08 is considered a good fit [66]. Additionally, the magnitude of the factor loadings (>0.50) [67] and the degree of factorial simplicity of an item were evaluated using the factor simplicity index (FSI) [68], establishing values greater than 0.70 as acceptable indicator criteria [69].

To evaluate SO_2_, the score reliability was estimated using the Cronbach’s alpha coefficient (α) [70], expecting magnitudes over 0.65 [71]; moreover, given that the α coefficient tends to be underestimated when the number of items is very low, the average inter-item correlation (*r_ij_* > 0.20) [72] was implemented as a complement. Additionally, the construct reliability was measured with McDonald’s omega (ω) [73], where coefficients over 0.70 are acceptable [74].

To comply with SO_3_, measurement invariance between men and women was analyzed with the selected model using a multiple group factor analysis, specifying consecutive and cumulative restrictions on the statistical equality of different parameters between groups [75], such as the total configurational equivalence of the PAS (configurational invariance), of the factor loadings (weak invariance), of the thresholds (strong invariance), and of the residuals (strict invariance). The degree of invariance was calculated by considering the variation in the magnitude of the CFI and RMSEA between nested models. Thus, there is unfavorable evidence for measurement invariance if ΔCFI < −0.01 and ΔTLI < −0.01 [76], and the ΔRMSEA ≥ 0.01 and ΔSRMR > 0.03 [77], or if ΔCFI < −0.002 and ΔRMSEA ≥ 0.007 [78]. The difference between men and women was calculated using Cohen’s d statistic: less than 0.41, negligible; between 0.41 and 1.15, low; between 1.15 and 2.70, moderate; above 2.70, high [79].

## 3. Results

### 3.1. Descriptive Analysis

Table 2 outlines the descriptive analysis (mean, standard deviation, skewness, and kurtosis) of the 12 items on the original scale. The highest mean was 5.95 (SD = 1.19) on item 7, “Some laws are not fair”. The lowest mean was 3.77 (SD = 1.34), corresponding to the item “The government laws and policies are effective”. The mean and SD of factors 1 and 2 were calculated based on the theoretical structures of 12 items.

### 3.2. Factor Structure

The goodness-of-fit indices are shown in Table 3, wherein the M4 ESEM model with seven items presents the best fit. One-factor (M1) and two-factor (M2) models were tested according to the original theoretical model, resulting in inadequate goodness-of-fit indices. The ESEM model applied to the original scale allowed the identification of low loadings, Table 4, in items 3 (People are cooperative), 4 (Most of the people think that honesty doesn’t work all the time, dishonesty is sometimes a better approach to get ahead), 7 (Some laws are not fair), 8 (Politicians don’t care about the problems of the average person), and 9 (The government laws and policies are effective).

These results confirm the two-factor correlated theoretical structure, although with a low magnitude (corr = 0.128; *p* = 0.034) proposed for the Perception of Anomie construct, it mostly presents adequate fit indices; however, it is observed that the RMSEA exceeds the limit considered as acceptable. However, the SRMR is the smallest, well below the acceptable threshold, which would indicate that the M4 model fits the data best.

### 3.3. Evidence of Reliability

Table 5 shows that, in terms of construct reliability, the factor breakdown of leadership shows favorable indicators, whereas the breakdown in social fabric factor is close to acceptable. By contrast, the evidence regarding the reliability of the scores is not uniform in terms of their indicators, since, while the α coefficient does not present acceptable magnitudes, the *r_ij_* does.

### 3.4. Factorial Invariance by Sex

Concerning the analysis of measurement invariance according to sex, the variation of fit indices (Table 6) indicates that the measurement of perceived anomie is measured equivalently between men and women considering the structure of the instrument (configural invariance), factor loadings (weak invariance), thresholds (strong invariance), and residuals (strict invariance). The mean of the items that make up factor 1 for men was 5.03 (SD = 0.89) and the mean for women was 4.93 (SD = 0.97), while for factor 2 it was 3.82 (SD = 1.38) and 3.77 (SD = 1.17), respectively, with no statistically significant differences found between men and women in both factors (d < 0.41).

## 4. Discussion

This study aimed to evaluate the factor structure, reliability, and measurement invariance of the Perception of Anomie Scale (PAS) in a sample of Chilean university students. First, the scale with the two-factor correlated structure was tested (“Breakdown in social fabric” and “Breakdown of leadership”), following the theoretical model and consistent with previous studies [15,24,27,32,33,34,35,45]. However, the model was fitted better in a population of university students from southern Chile by eliminating five items from the original version (3, 4, 7, 8, and 9).

There are several reasons why these items did not perform as expected; perhaps there were problems in understanding the items [80], or perhaps it was due to the presence of inverse items [81]. In the dimension “Breakdown in social fabric”, comprising the first six items, only item 3 was written in the positive; in the dimension “Breakdown of leadership”, comprising items 7 to 12, only items 7 and 8 were in the negative, confirming that the items within this dimension that were contrary to the polarity of the majority did not load adequately. Items 4 and 9 were eliminated because they were below the cut-off criterion (<0.5). In addition, the goodness-of-fit indices improved with their elimination. It is important to note that eliminating these items did not affect the theoretical structure of the original construct. The dimension breakdown in social fabric contained items addressing trust (1 and 2) and moral decline (5 and 6); the same was true for the dimension “Breakdown of leadership”, which maintained items on legitimacy (11 and 12) and effectiveness (10). Additionally, while the correlation between factors is relatively low, it can be explained by the combination of methods used (ESEM and factorial simplicity), because this eliminated items that were significantly influenced by more than one factor, and consequently, artificially raised inter-factor correlations. Finally, it is necessary to mention that there are likely other structural reasons (e.g., correlation between residuals) that explain the behavior of the RMSEA in the last model, since it is more sensitive to misspecifications than other indicators [82] and to sample size [83].

The study closest to the objectives of this study was conducted in a Colombian university population, where the validity of this scale was reviewed [45], and items 1 and 12 were eliminated, considering only an internal consistency analysis through Cronbach’s α. Reliability for the dimension “Breakdown of leadership” was adequate; however, it was weak for the dimension “Breakdown in social fabric”, which is consistent with previous studies employing a university population [15,45]. While the low reliability reported for the “Breakdown in social fabric” dimension might be concerning, in a basic research context, it could be tolerable, given that its factor loadings are acceptable (>0.50 [67]), implying an adequate representation of the construct in studies analyzing latent variables. However, it is necessary to cautiously interpret the results of these applications and take actions to optimize the measure. On the other hand, it would not be advisable for use in applied research because it is typically focused on decision-making about individuals.

The factorial invariance of the PAS as a function of sex was examined. The results show that the PAS measured men and women equivalently. This included factor loadings (weak invariance), thresholds (strong invariance), and residuals (strict invariance). These findings are relevant because we were able to compare the means between men and women. In that regard, after conducting that procedure, no significant differences were found between these groups, a finding that aligns with the study by Teymoori et al. [15], who made comparisons of means between men and women and found no significant differences, although without performing a previous analysis that would enable an equitable comparison. The evidence indicates that the Chilean sociopolitical context and recent events such as the political and economic crisis, and the pandemic outbreak, cause anomie to be perceived equally by both men and women.

In relation to the conceptual contributions of this study, as noted, there are two main currents in the study of anomie. One is linked to more psychological aspects, related to mental processes of people living in an anomic society, a line that addresses, among others, deviant behaviors, crime, despair, personal confusion, frustration, meaninglessness, powerlessness, and loneliness [16,17,18,19,20]. The other is the more social current, to which this study subscribes, linked to the initial developments made by Durkheim [3] on social integration and regulation, but which in the current discussion incorporates elements of intersubjectivity to assess society [15]. This approach makes it possible to see anomie as a social construct that may or may not be influenced by objective events a community or country experiences [15]. Perhaps more importantly, however, this intersubjective construction of society influences key aspects of today’s society. For example, at the psychological level, it may help to explain mental health issues [84]; at the social level, it may be a key variable in understanding social discontent [21,22], social mobilizations, and violent extremism [23]. At the economic level, there is evidence of its close relationship with inequality [24,26], and at the political level it could help explain the political fragmentation and pendulum-swings to the extremes [24,26,27,32].

The Perception of Anomie Scale provides a valuable measure of the state of Chilean society through the lens of university students who have always pushed for social demands [85]. In applying the original study, Chile averaged 4.53 on the PAS, ranking 13th out of 28 countries with more anomie; in this study, the average increased to 4.6. There is a perception among university students that the country is anomic [86], and this could explain the mental health problems [87], the demands for greater equality that occurred during the social outburst of October 2019, and the political fragmentation and growing polarization of Chilean society today. Certainly, the Chilean social context has influenced the increase in the anomie perception. This statement aligns with Teymoori et al. [15], who indicated that most stable countries have a lower perception of anomie, while those experiencing economic crises, social changes and internal conflicts show a higher perception of anomie.

One of the study’s main limitations is related to the representativeness of the sample. Only students from universities located in Temuco participated, mainly from the Araucanía region, a territory that represents about 6% of the national population. Furthermore, there was an insufficient number of individuals who identified themselves as non-binary to establish differences in the latent structure of anomie. In the future, it will be necessary to work with probability samples representing other age groups and cities. Likewise, it could be useful to conduct a more thorough review of the contents of items in the factors that present lower reliability coefficients to detect those aspects that can be improved, whether in relation to the evaluated construct or the wording of the item. In addition, evidence of metric invariance should be provided for different subgroups, such as indigenous populations, socioeconomic strata, and political tendencies, as well as a validity analysis for their relationship with other theoretically relevant variables.

## Figures and Tables

**Figure 1 behavsci-14-00172-f001:**
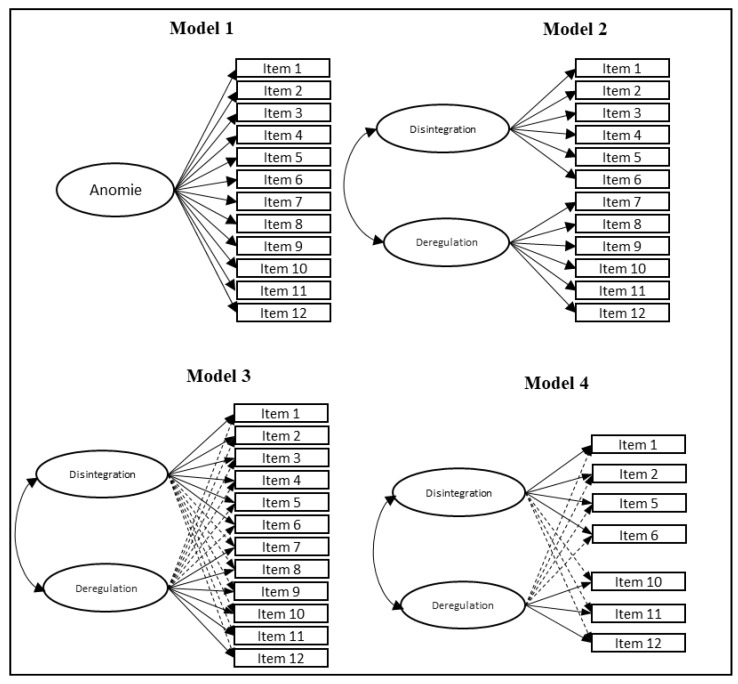
Models for PAS.

**Table 1 behavsci-14-00172-t001:** Perception of Anomie Scale (PAS).

Item	Content
	Instruction: Think of Chile society and indicate to what extent do you agree whit the following statements? In Chile today…
Item 1	People do not know who they can trust and rely on
Item 2	Everyone thinks of himself/herself and does not help others in need
Item 3	People are cooperative
Item 4	Most of the people think that honesty doesn’t work all the time; dishonesty is sometimes a better approach to get ahead
Item 5	People think that there are no clear moral standards to follow
Item 6	Most of people think that if something Works, it doesn´t really matter whether it is left or wrong
Item 7	Some laws are not fair
Item 8	Politicians don´t care about the problems of average person
Item 9	The government laws and policies are effective
Item 10	The government works towards the welfare of people
Item 11	The government is legitimate
Item 12	The government uses its power legitimately

**Table 2 behavsci-14-00172-t002:** Descriptive statistics.

Item	M	SD	g1	g2
Item 1	5.48	1.38	−1.27	1.71
Item 2	5.00	1.41	−0.68	0.19
Item 3	4.26	1.27	−0.74	0.20
Item 4	4.23	1.77	−0.35	−0.90
Item 5	4.38	1.47	−0.40	−0.39
Item 6	4.99	1.47	−0.79	0.21
Item 7	5.95	1.19	−1.36	1.88
Item 8	5.67	1.25	−1.06	1.12
Item 9	3.77	1.34	−0.34	−0.40
Item 10	4.05	1.39	−0.45	−0.58
Item 11	4.41	1.61	−0.13	−0.65
Item 12	4.15	1.46	−0.07	−0.40
F1_total_	4.72	0.77	−0.42	0.65
F2_total_	4.67	0.77	−0.36	0.12

Notes. Mean (M), standard deviation (SD), skewness (g1), kurtosis (g2).

**Table 3 behavsci-14-00172-t003:** Goodness-of-fit indicators.

Models	χ^2^ (gl)	CFI	TLI	RMSEA (CI 90%)	SRMR
M1: AFC One-factor	650.058 (54) *	0.669	0.596	0.170 (0.158–0.182)	0.092
M2: AFC Two-factor	350.251 (53) *	0.835	0.795	0.121 (0.109–0.133)	0.066
M3: ESEM Two-factor 12 item	222.351 (43) *	0.900	0.847	0.104 (0.091–0.118)	0.045
M4: ESEM Two-factor 7 item	33.377 (8) *	0.982	0.953	0.091 (0.060–0.124)	0.020

Notes. CFI = comparative fit index; TLI = Tucker–Lewis index; RMSEA = root mean square error of approximation; SRMR = standardized root mean square residual. Estimator: WLSMV. * *p* < 0.001.

**Table 4 behavsci-14-00172-t004:** Standardized factor loadings. ESEM two-factor model.

	ESEM n = 383					
	12 Item			7 Item		
Items	F1	F2	FSI	F1	F2	FSI
Item 1	**0.48**	0.07	0.98	**0.58**	0.05	0.99
Item 2	**0.52**	0.03	1.00	**0.56**	−0.01	1.00
Item 3	0.23	0.16	0.67			
Item 4	0.43	−0.08	0.96			
Item 5	**0.50**	0.01	1.00	**0.53**	0.00	1.00
Item 6	**0.59**	0.00	1.00	**0.53**	−0.03	1.00
Item 7	0.31	0.13	0.83			
Item 8	0.37	0.14	0.85			
Item 9	0.19	0.42	0.81			
Item 10	0.20	**0.67**	0.90	0.19	**0.60**	0.89
Item 11	−0.22	**0.88**	0.93	−0.13	**0.89**	0.98
Item 12	−0.13	**0.79**	0.97	0.00	**0.80**	1.00

Notes. F1: Breakdown in social fabric. F2: Breakdown of leadership. FSI = factor simplicity index. Values in bold highlight the highest factor loadings.

**Table 5 behavsci-14-00172-t005:** Reliability indicators PAS (7 items).

	ω McDonald	α Cronbach	*r_ij_*
Breakdown in social fabric	0.566	0.566	0.246
Breakdown of leadership	0.800	0.785	0.549

Notes. *r_ij_* = average inter-item correlation.

**Table 6 behavsci-14-00172-t006:** PAS measurement invariance.

Model	χ^2^	df	CFI	TLI	RMSEA	90%CI	SRMR	ΔCFI	ΔTLI	ΔRMSEA	ΔSRMR
1. Configural invariance	58.588	35	0.984	0.981	0.060	0.031–0.086	0.025				
2. Weak invariance	77.020	45	0.978	0.979	0.062	0.037–0.084	0.033	−0.006	−0.002	0.002	0.008
3. Strong invariance	90.915	59	0.978	0.984	0.054	0.030–0.075	0.032	0.000	0.005	−0.008	−0.001
4. Strict invariance	119.114	66	0.963	0.976	0.066	0.047–0.084	0.036	−0.015	−0.008	0.012	0.004

## Data Availability

The raw data supporting the conclusions of this article will be made available by the authors on request.

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
