# Peer review of "Adaptation and Validation of the Perception of Anomie Scale in Chilean University Students"

_behavsci, 2024, doi:10.3390/bs14030172_

Round 1

Reviewer 1 Report

Comments and Suggestions for Authors

Author Response

Dear reviewer:

We greatly appreciate your comments and contributions to improve the manuscript. The suggestions were considered and fully addressed.

Reviewer 2 Report

Comments and Suggestions for Authors

Comments to the authors:

The reviewer thoroughly examined the submitted paper entitled "Adaptation and Validation of the Perception of Anomie Scale in Chilean University Students" with keen interest. In this paper, the authors conducted an investigation into the validity of the Perception of Anomie Scale (PAS) by implementing an experiment involving Chilean university students. Consequently, they confirmed the validity of PAS, characterized by two factors (disintegration and deregulation), and elucidated the invariance in PAS between females and males.

The reviewer acknowledges the potential contribution of the submitted paper to the field of anomie studies. However, some concerns have arisen regarding the statistical models employed in the authors' study and the interpretation of their findings. While the reviewer believes that these concerns do not significantly diminish the significance of the authors' study, it would be desirable for them to address these issues in the revised paper. Upon the satisfactory resolution of these concerns, the reviewer intends to recommend the paper for publication.

Firstly, the authors did not conduct a statistical examination of the correlation between the two factors (disintegration and deregulation). In other words, the authors implicitly assume that these factors are independent of each other. However, the reviewer finds this assumption unacceptable. Additionally, the authors did not examine Model 2 when limited to 7 items, as opposed to 12 items. By considering the correlation between the two factors, the reviewer suggests that there is a potential for acquiring a more effective model than the ones utilized by the authors. If true, the authors may uncover additional interesting facts related to the latent structure of anomie.

Secondly, as one of the reasons the authors focused on Chilean university students was the claim that Chilean society is an anomic society, the reviewer is interested in understanding how the results of the authors' study relate to such characteristics of Chilean society. In other words, the reviewer believes that the authors should theoretically discuss the effects of social contexts on the latent structure of anomie found in Chilean society.

Thirdly, the authors reported the inclusion of individuals with non-binary gender in the participant pool. However, the authors only examined differences in the latent structure of anomie between females and males. While the number of non-binary individuals in the study is extremely small, the reviewer does not believe that additional analysis for non-binary gender is necessary. However, considering the possibility that the latent structure of anomie found in females and males may not apply equivalently to the non-binary gender group, the reviewer suggests that the authors acknowledge this as a limitation of their study.

Finally, it appears that two "DE" entries in the first line of Table 2 might be typos and should be "SD." The reviewer recommends verifying this.

Thank you very much for providing the opportunity to review the paper. The reviewer greatly appreciates the chance to contribute comments aimed at improving the paper.

Author Response

(The authors gave the same response as above.)
